# Incremental Clustering for Predictive Maintenance in Cryogenics for Radio Astronomy

**DOI:** 10.3390/s24072278

**Published:** 2024-04-03

**Authors:** Alessandro Cabras, Pierluigi Ortu, Tonino Pisanu, Paolo Maxia, Roberto Caocci

**Affiliations:** National Institute for Astrophysics (INAF), Cagliari Astronomical Observatory, Via della Scienza 5, 09047 Selargius, Italy; pierluigi.ortu@inaf.it (P.O.); tonino.pisanu@inaf.it (T.P.); paolo.maxia@inaf.it (P.M.); roberto.caocci@inaf.it (R.C.)

**Keywords:** predictive maintenance, cryogenics, radio astronomy, anomaly detection, unsupervised machine learning

## Abstract

In a cooling system for radio astronomy receivers, maintaining cold heads and compressors is essential for consistent performance. This project focuses on monitoring the power currents of the cold head’s motor to address potential mechanical deterioration, which could jeopardize the overall functionality of the system. Using Hall effect sensors, a microcontroller-based electronic board, and artificial intelligence, the system detects and predicts anomalies. The model operates using an unsupervised approach based on incremental clustering. Since potential fault scenarios can be multiple and often challenging to simulate or identify during training, the system is initially trained using known operational categories. Over time, the system adapts and evolves by incorporating new data, which can be assigned to existing categories or, in the case of new anomalies, form new categories. This incremental approach enables the system to enhance its performance over the years, adapting to new anomaly scenarios and ensuring precise and reliable monitoring of the cold head’s health.

## 1. Introduction

A radio telescope is a complex structure designed to receive and measure the Radio Frequency (RF) power emitted by radio astronomical sources. The received signal is essentially a broad-band noise with statistical properties similar to both the background noise and the noise generated by the radio telescope’s receiving system itself. Power levels of radio astronomical signals are typically in the order of $10^{-15}÷10^{-20}$ W; for this reason, one of the fundamental requirements for the proper functioning of receivers installed in a radio telescope is sensitivity.

The total noise power available from the “radio telescope system” can be expressed by the following expression [1,2]: (1)$$P_{sys}=P_{A}+P_{rec},$$

In Equation (Equation 1), $P_{A}$ represents the available noise power at the output terminals of the antenna, while $P_{rec}$ is the receiver noise power. Considering the power–temperature equivalence, expressed by Equation P=kTB (where k is the Boltzmann constant, T is the temperature in Kelvin, and B is the bandwidth of the received signal), the expression (1) can be written as: (2)$$T_{sys}=T_{A}+T_{rec},$$
where $T_{A}$ represents the sky noise temperature of the antenna [3] and depends on the pointing direction (elevation) of the antenna: typically it ranges from 25 K for elevation angles of 45° to 90 K for elevation angles of 5° (or near the Earth’s horizon). $T_{rec}$ indicates the receiver temperature and is determined by the temperatures of various components of the receiving chain. The $T_{sys}$ mentioned in Equation (Equation 2) is related to the sensitivity from the radiometer Equation [2]: (3)$$\frac{S}{N}=\frac{T_{src}\sqrt{\Delta_{f}\tau }}{T_{sys}},$$

In Equation (Equation 3), *S/N* is the SNR (Signal-to-Noise Ratio), $T_{src}$ represents the temperature of the observed radio source, and τ is the integration (or observation) time. Generally, because $T_{sys}>>T_{src}$, the reduction in contributions due to $T_{A}$ and $T_{rec}$ becomes very important. Neglecting the description of techniques utilized for the reduction of the antenna temperature $T_{A}$, in this context it is crucial to consider that the receiver noise temperature $T_{rec}$ depends on the number of components in the receiving chain. Its value can be quantified using the Friis formula [4,5], which states that the equivalent noise temperature of a generic system composed of *N* devices is equal to [6]: (4)$$T_{tot}=T_{1}+\frac{T_{2}}{G_{1}}+\frac{T_{3}}{G_{1}G_{2}}+\ldots +\frac{T_{N}}{G_{1}G_{2}\ldots G_{\left(N-1\right)}},$$
where $T_{i}$, $G_{i}$ (with i=1,…,N) are, respectively, the equivalent noise temperatures and gains associated with individual components of the receiving chain. Since the noise temperature of these components depends on their physical temperature, cooling them to extremely low temperatures near absolute zero reduces the thermal agitation of the electronic components. Therefore, ensuring the cooling of the entire chain is of fundamental importance, as described, for example, in [7,8], in the development of LNAs for radio astronomy.

To achieve the required temperatures, the components of the receiving chain are positioned inside cryostats (Dewars). Typically, they consist of two stages: the first operates at temperatures around 70 K, while the second, used to reach lower temperatures, operates around 10 ÷ 20 K. The most crucial feature of a cryostat is the refrigeration capacity and the minimization of thermal load: this aspect is related to heat conduction mechanisms (radiation, conduction, and convection) [6]. The reduction contribution of the radiation is achieved using special materials such as super-insulation, while the conduction is reduced by minimizing the thermal link between the parts of the cryostat cooled to 70 K and 20 K. The convection is significantly reduced by creating (and maintaining) a vacuum inside the dewar: removing air prevents thermal exchange caused by air currents generated when adjacent surfaces have different temperatures. In general, a cryogenic dewar can be formed by a temperature- and vacuum-level monitoring system (vacuum gauge), a vacuum generation system (whose main component is the vacuum pump), a refrigeration circuit including a cold head and the gas compressor used to achieve low temperatures (usually helium), along with the necessary components for the electrical power supply.

For the reasons mentioned above, the maintenance of cold heads and compressors is crucial for sustaining the required performance over time.

This work aims to monitor the power currents that supply the cold head motor as its mechanical components are subject to wear and can compromise the operation of the entire system. Any anomalies in the currents can allow for the early prediction of potential malfunctions, reducing unplanned downtime.

### State of the Art

Unsupervised machine learning [9] has significant applications across various sectors. In the business context, clustering is crucial for identifying customers based on similar purchasing behaviors, enabling companies to tailor marketing strategies more precisely. In social networks, the use of clustering facilitates connections between users sharing similar preferences, enhancing the overall user experience. In the realm of predictive maintenance, clustering proves useful in applications where anomalies are unlabeled in the initial project phase [10] or in continual learning scenarios where the number of classes cannot be defined a priori [11]. Currently, in cutting-edge research, numerous studies compare the performance of various clustering algorithms, providing valuable insights for selecting the most suitable method for specific needs [12].

In the context of the problem examined in this research, the application of unsupervised algorithms proves particularly fitting for the outlined use case. In the following sections, the process for data collection and model training will be detailed.

## 2. Materials and Methods

### 2.1. Hardware Description

The monitoring system has been designed to ensure that any potential malfunction does not compromise the system’s operability and does not adversely affect its MTBF (Mean Time Between Failure). This is guaranteed by the fact that the three power supplies required for the three-phase motor enter the box and are intercepted without undergoing changes or alterations from the signal processing electronics. Therefore, the prototype implemented is a pass-through system. Figure 1 illustrates the integration of the Cold Head Monitor (CHM) into the refrigeration system composed of the compressor and the cold head.

Examining the control system in detail in Figure 2, several macro-blocks can be identified:The first board, named HALL V1, contains Hall effect sensors ACS758xCB, produced by Allegro Microsystems, Manchester, NH, USA [13].The second board, named HALL CONDITIONING V1, includes the amplification stage and the voltage acquisition with the Analog-to-Digital Converter (ADC) ADS1015, produced by Texas Instruments, Dallas, TX, USA [14].An Arduino LEONARDO produced by Interaction Design Institute, Ivrea, Italy [15] for logic and additional acquisition through digital pins.A 230 VAC to 5 VDC power supply for powering the sensors, ADC, and the Arduino itself.

**Figure 2 sensors-24-02278-f002:**
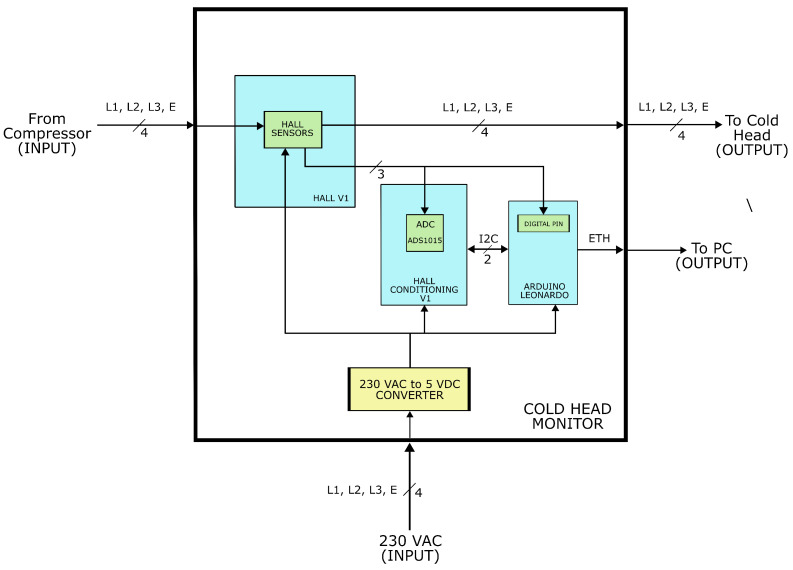
Internal description of the CMH system: the blue blocks represent the Hall effect sensors, signal conditioning, signal acquisition, and data processing, while the yellow ones indicate the electrical power supply.

From the CTI 9600 compressor [16], in addition to the helium lines that go directly to the cold head (not shown in Figure 2), power supplies (L1, L2, L3) along with the ground (E) cables are delivered. L1, L2, and L3 are intercepted by the HALL V1 board, which houses the Hall effect sensors, before being output without alteration.

The voltage outputs from these sensors are extracted and directed to the HALL CONDITIONING V1 board, equipped with a 12-bit ADC from Adafruit. At the same time, they are routed to the Arduino Leonardo, on the Digital Pins 0, 1, and 7, which are connected to timers.

The parallel acquisition of the data is motivated by the need to sample the amplitudes of the voltages proportional to the 3 currents with the first ADC, while the respective phase shifts are captured with the Digital Pins.

These three boards (Hall sensors, ADC, Arduino) need to be powered, hence the presence of an AC–DC Converter. Finally, signal visualization, storage, and any post-processing occur through a PC, which acquires data via an Ethernet protocol.

### 2.2. Firmware 1.0v and Acquisition Software 1.0v

Dedicated firmware written in C has been developed to handle communication with the Arduino microcontroller, managing Ethernet communication and providing a set of APIs to query the device for receiving the three data streams. Specifically, gains are calculated for the three channels of the ADC, enabling an assessment of the signal amplification. Additionally, phase shifts between signals in channels 1–2 and 1–3 are determined using the Arduino interrupts, providing insights into the temporal delay between the signals. The values obtained from the ADC for the three channels are expressed in peak-to-peak volts, allowing the amplitude measurement of the signals. The collected data are visualized through a simple software interface written in Python, which offers a variety of information about the system’s condition. This interface not only facilitates the visualization of the mentioned information but also allows real-time plotting of the current, along with the capability to save and load temporal sequences. The software allows for visualizing the health status of the cryogenic heads by integrating the module described in the following section.

### 2.3. Clustering Setup

In addition to these quantitative analyses, the data undergoes a more advanced evaluation process through an artificial intelligence system. The training dataset was created by collecting data from cold head tests with various operating hours and different uses. As depicted in Table 1, our study employed two types of cold heads provided by CTI-Cryogenics [17], which differ solely in their cooling capacity rather than their electrical characteristics. We chose to adopt the 30,000 h threshold as a criterion for maintenance, aligning with the manufacturer’s recommendations. This time interval was selected as a reference point to distinguish between normal operation and maintenance states. Both types of heads were alternately mounted inside an empty test Dewar, and powered using the same compressor. This approach allowed us to maintain consistent experimental conditions during data acquisition. The three data streams from the Arduino (one for each channel) were divided into 128-sample lengths to obtain data suitable for clustering analysis. This sample size enables the generation of data easily processable by the clustering model and includes sufficient information to distinguish the samples themselves.

The data preprocessing stage involved a series of transformations aimed at optimizing the data for subsequent analysis. In particular, the initial dataset, consisting of sinusoidal signal windows with a length of 128 samples (as illustrated in Figure 3), underwent a process of statistical synthesis. This process involved the extraction of key statistical indicators from each window, resulting in a significant dimensionality reduction. Specifically, the preprocessing function computed five fundamental statistical features for each window: the mean, standard deviation, maximum value, minimum value, and range (i.e., the difference between the maximum and minimum values). By condensing the information within each window into these concise statistical summaries, the computational burden on the subsequent clustering algorithm was effectively reduced. These features provide information about the distribution and amplitude of the signal within the window.

The model operates through an unsupervised approach based on incremental clustering [18], a technique used in data mining and machine learning to update a clustering model with new data without having to rebuild the entire model from scratch. It adjusts existing clusters or creates new ones as new data arrive, allowing for efficient and scalable updates to the model. Initially, the system is trained considering three types of operation: normal operation, a cold head requiring maintenance, and a cold head with a non-functioning compressor. These categories were identified through the analysis and availability of the collected data; however, it is understood that fault scenarios can be diverse, complex, and often challenging to simulate or identify during training.

The detection of new anomalies is performed by analyzing new input data in real time within the system. These new data points can be assigned to one of the three existing clusters or, in the case where the anomaly is different from the labeled ones, contribute to forming a new cluster.

The assignment of a point to a cluster is determined by calculating the Euclidean distance between the point and the centroids of the clusters. The point is then assigned to the cluster with the closest centroid. Formally, given a set of points *X* and *k* centroids $X=\{c_{1},c_{2},...,c_{k}\}$, the point is assigned to the cluster with the nearest centroid.
(5)$${\mathrm{argmin}}_{j}{∥x_{i}-c_{j}∥}^{2}$$
where ‖·‖ represents the Euclidean norm. To identify a point as anomalous, a threshold δ is considered on the Euclidean distance. If the distance between the point and the nearest centroid is greater than δ, the point is considered anomalous. Formally: (6)$$∥x_{i}-c_{{\mathrm{argmin}}_{j}}∥>\delta$$

The value of δ has been selected by considering the distribution of distances between all points in the dataset and the cluster centers of the trained model. The choice of an upper value falls within a specific confidence interval.

This incremental approach allows the system to autonomously improve over the years, adapting to new anomaly scenarios and ensuring precise and reliable detection of the health status of cold heads, even in the presence of previously unidentified anomalies. However, it is important to emphasize that, at the current state of the system, retraining the model is necessary in the presence of anomalies.

The choice of the algorithm used was influenced by various factors. K-means was chosen as the solution because it offers numerous advantages over other clustering algorithms, such as DBSCAN and OPTICS, especially in contexts where the number of clusters is known a priori. Its superior computational efficiency is attributed to its conceptual simplicity and fast convergence as an iterative algorithm [19].

Furthermore, K-means stands out for its ease in handling a predefined number of clusters, ensuring a more straightforward and controllable implementation compared to algorithms like DBSCAN, which rely on parameters such as density and distance.

Another crucial point is the availability of the predict function in K-means, enabling inferences on new data by assigning them to existing clusters. This feature proves particularly useful in continuous monitoring scenarios, where real-time data classification is essential [20].

## 3. Results and Discussion

The K-means algorithm, trained with the dataset, revealed a distinct separation between the three classes identified in the project. In Figure 4, the scatter plot for two of the five features used is presented, while Table 2 lists the most important key evaluation metrics for clustering:The Silhouette score [21] provides a measure of internal cohesion and separation between clusters. The score ranges from −1 to 1, where a higher score indicates better clusters.The Calinski–Harabasz index [22] measures the cluster’s compactness. A higher value indicates better clusters.The Davies–Bouldin index [23] measures the “compactness” and “separation” of clusters, with lower values indicating better clusters.

**Table 2 sensors-24-02278-t002:** Common evaluation metrics obtained from training with the K-means algorithm.

Metric	Value	Range
Silhouette score	0.7848	[−1, 1]
Calinski–Harabasz index	$8.1762\times 10^{6}$	[0, +∞]
Davies–Bouldin index	0.2163	[0, +∞]
Adjusted Rand index	98%	[0, 100]

**Figure 4 sensors-24-02278-f004:**
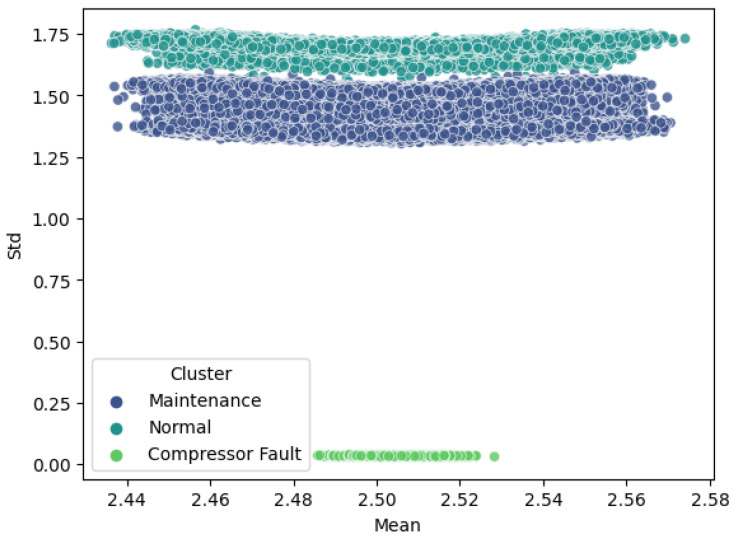
2D scatter plot of clusters obtained using the k-means algorithm considering two of the five features used: mean and standard deviation.

Moreover, due to its access to ground truth labels, the adjusted Rand index [24] is used to compare predicted labels with the actual ones.

The chosen value of δ to identify points as anomalies is 0.5 higher than the distances between the points and the centroids of the known clusters, as depicted in Figure 5. The model was trained on a workstation with 2 × EPYC 7402 24-core produced by AMD, Sunnyvale, CA, USA Processors, 256 GB of RAM and 2 × GeForce RTX 4090 GPUs produced by Nvidia, Santa Clara, CA, USA. In support of the findings in Section 2.3, a comparison was conducted between the K-means and DBSCAN algorithms in terms of training time across varying database sizes. The results presented in Figure 6 show that, in the case of the density-based algorithm, the training time increases linearly with the dataset size, while for K-means, it remains constant. This highlights its unsuitability for incremental applications [25,26]. The final system was tested, with data processed by the CHM sent over the network to an NVIDIA Jetson TX2, a powerful embedded computing platform [27] running the acquisition software. Using this configuration, the average inference time to assign correct labels to the current sample is approximately 0.03 s.

## 4. Conclusions and Future Works

The developed system stands out as a versatile solution for monitoring cryogenic systems, especially within the realm of radio astronomy. It adeptly discerns maintenance needs by assessing actual wear and tear, thereby reducing unnecessary maintenance intervals and optimizing maintenance scheduling. Its compact design makes it suitable for remote installation even in confined spaces, requiring limited resources. Additionally, its pass-through nature ensures that malfunctions do not compromise the entire system. Moving forward, future project developments will prioritize the integration of additional sensors (such as those for monitoring temperature and vibrations) and accounting for the phase difference among the signals from the three channels. Another aspect to consider will be automating the process of identifying new anomalies, aiming to eliminate the need for manual model training each time a new cluster is detected. It is worth noting that the prototype is well-suited to be applied to other types of motors. This requires minor hardware modifications and retraining of the AI model. Therefore, in the future, the potential extension of the prototype to other devices requiring predictive maintenance will be assessed.

## Figures and Tables

**Figure 1 sensors-24-02278-f001:**
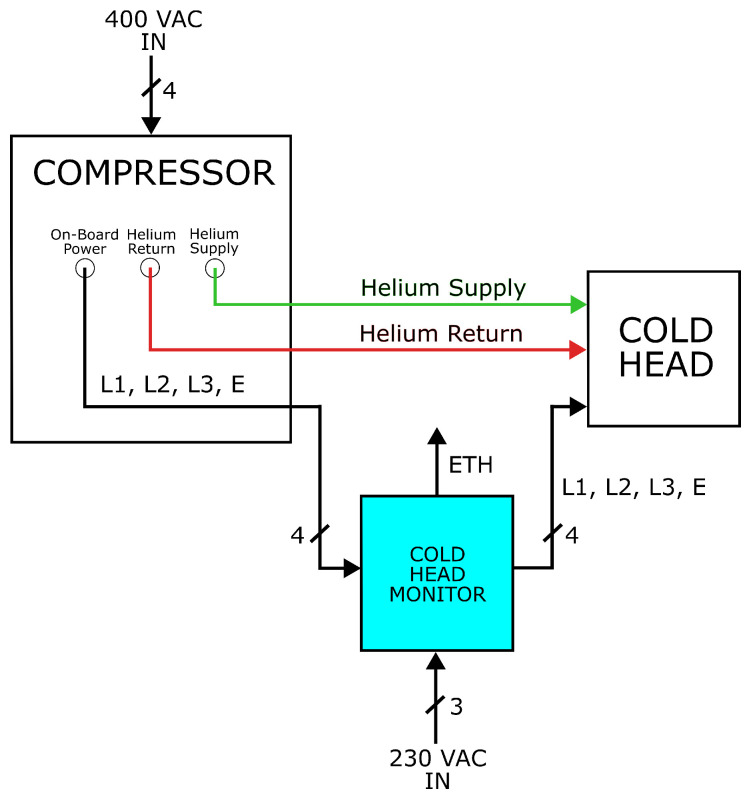
The block diagram depicts the system delivering both helium and electrical power from the compressor to the cold head, with the CHM device inserted in between.

**Figure 3 sensors-24-02278-f003:**
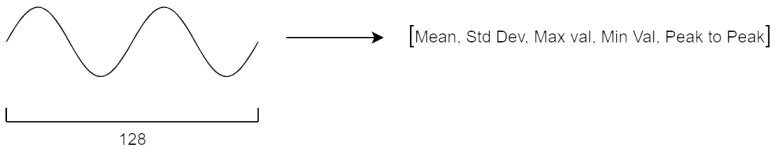
Features extraction from the windows of the signal.

**Figure 5 sensors-24-02278-f005:**
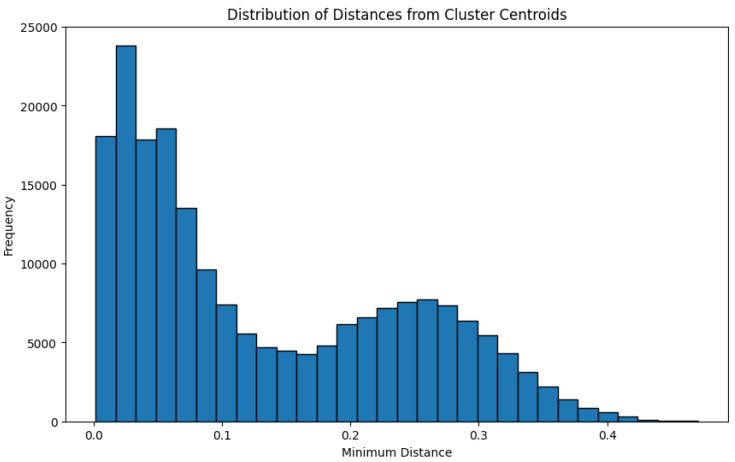
Minimum distance for each point in the dataset from the three cluster centroids.

**Figure 6 sensors-24-02278-f006:**
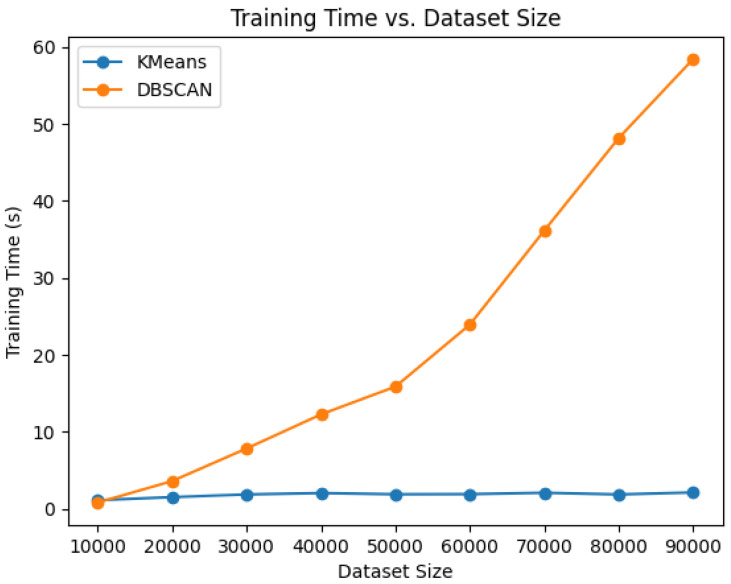
Training time comparison between K-means and DBSCAN clustering algorithms across dataset sizes ranging from 10,000 to 90,000 samples.

**Table 1 sensors-24-02278-t001:** Dataset composition: Model of cold head used, operating hours prior to data acquisition, samples collected for each channel, and class membership.

	Model	Operating Hours	Samples per Channel	Class
1	CTI-1020	>30 k	161,172	Maintenance
2	CTI-350	<30 k	139,098	Normal Status
3	CTI-350	>30 k	151,192	Maintenance
4	CTI-350	>30 k	130,716	Maintenance
5	CTI-350	<30 k	150,415	Normal Status
6	-	-	150,919	Compressor Fault

## Data Availability

The data presented in this study are available on request from the corresponding author.

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
