# Peer review of "Incremental Clustering for Predictive Maintenance in Cryogenics for Radio Astronomy"

_sensors, 2024, doi:10.3390/s24072278_

Round 1

Reviewer 1 Report

Comments and Suggestions for Authors

The authors of the paper addressing a technique to monitor the maintenance of cryogenic equipment in radio astronomy instrument. A couple of remarks:

- Although I understand the need of cryogenics for radio astronomy instruments, the implemented monitoring technique can be used in a broader sense. Why limit to RA? 

- I think the real strength of the method is the predictive aspect of it. However, I'm missing this in the paper. Did you analyse trends in the data, which can give information for future failures ? For cryogenic equipment it's always a trade-off between continuing with operations and maintenance. Changing cold heads and setting up the system is a time-consuming effort, with considerable down-time. 

- The paper is more an implementation of known techniques , rather than a new research topic. What's really new ? 

- The details of the implementation are missing. How do you map the measured data to the algorithm? What is the connection to c and ? More detail is needed here.

- On Page 5 the signal window length of 128 is given. More info is needed here. What is the data, 128 what ? 

- The scores mentioned on Page 6 are interesting, but more info is needed. 

- Three groups in Figure 4 is interesting, but what's really interesting is signs for predictive maintenance. Please elaborate on that in the paper.

- In the conclusions I'm missing some conclusions on the used classification and decision techniques. 

Comments on the Quality of English Language

Well written paper. IMHO the references to equations should be with a capital as well (... Equation 4 ..) 

Author Response

Thank you for your review. Below, I address each of the corrections you have suggested, hoping to be sufficiently thorough. It is important to emphasize that the article falls under the category of "communication" and represents only the first part of a larger project.

  • The project focuses on radio astronomy, from the need to monitor the health status of the cold heads at our research institute, which are essential for cooling the receivers. Furthermore, such an application in this field is, in our opinion, lacking in the current state of the art. However, as highlighted in the conclusions, a system of this kind can also be successfully extended to other applications.
  • Currently, the cold heads are sent for maintenance after accumulating 30,000 hours of operation, regardless of their actual wear and tear. However, by using a system of this kind and training it with cold heads having different numbers of operating hours, it is possible to identify critical wear and intervene with maintenance only when necessary, thereby optimizing such operations.
  • We have provided additional details and justified the use of 128-sample windows. Samples are expressed in peak-to-peak volts, as explained in Section 2.2.

  • The metrics listed in Table 2 are commonly used in clustering and are aimed at comparing two similar systems to assess their quality. To ensure a better understanding for the reader, we have included some relevant bibliographic references.
  • The clusters depicted in Figure 4 correspond to the 3 operating conditions we have identified. We collected samples using cold heads with less than 30.000 hours of operation, indicating they do not require maintenance according to the manufacturer's guidelines. Additionally, we included samples from cold heads with more than 30.000 hours of operation, indicating they require maintenance. Furthermore, we included instances of the compressor being off, as it can serve as an important alert for our system. The clustering algorithm, described in reference 12, successfully recognizes and distinguishes between these 3 scenarios, as depicted in Figure 4.

Please note that modifications to the text are highlighted in bold. Image captions have been improved, and syntax errors have been corrected. Additionally, more details have been added to the conclusions.

Reviewer 2 Report

Comments and Suggestions for Authors

This paper addresses a potentially interesting issue regarding the maintenance of cryogenics systems used in radio astronomy. The authors claim that by using Hall sensors  to probe the current circulating into the 3 phases of the motor running the cold head of a cryocooler it is possible to predict the scheduling of maintenance. This is achieved by using an unsupervised approach based on incremental clustering.

This paper has several problems. 

Abstract: The abstract opens with a statement about compressors and cryopumps while the paper is about the motor presumably operating a piston in a G-M cooler. In addition, cryopumps are not usually part of cryogenic systems for radio astronomy. 

Introduction:  This introduction is completely out of context. It is a brief and elementary description of a radio telescope with mention to noise in (presumably) low noise radio receivers. Then mechanisms of thermal conductions inside cryogenic systems are described. The only weak link with the rest of the paper is in the need of maintenance of compressors and cryopumps (again). Unfortunately their electronics is monitoring only the 3 phases going to the G-M cold head and not any parameters involving the compressor. The cryopumps are again mentioned but they are not commonly used - to the best of my knowledge - in cryogenic receivers.

in 1.1 a state of the art is described in the contect of machine learning for business, marketing strategies and social networks which is completely out of contect in a technical paper about maintenance of a cryoenic system used in radio astronomy..

Materials and methods

This section briefly describes the electronics used to probe the 3 phases feeding a G-M cold head. No where a description of the combination compressor-cold head is given. I inferred that 3 hall sensors probe the 3 phases going to the G-M cold head. The signal from the hall sensor is amplified and then digitized before being fed to an arduino Leonardo board.

Software setup. This section describes - without any details - that some software has been written to process the 3 real time data stream. I understand that the amplitude and phase differences are obtained.

Clustering setup. This section should be the core novelty of this paper. It briefly mention that the data undergo a “more advanced evaluation process through an artificial intelligence system”.  Then a sinusoidal signal (one of the 3 phases?) is analyzed by extracting mean, std dev, max, min and delta. The notion of incremental clustering is mentioned but nowhere I could find any description of what is clustered. I understand that data is collected when a system (what system?) is running normally, needs maintenace and faulty. The process of data collection for these three conditions is not described. The AI software will calculate a “distance” between the real time data and each of the three clusters with an incremental approach. Finally, some mention to some algorythms (K-means, dbscan, optics) is given.

Results and discussion. In this section it seems that the authors declare that their algorythm is K-MEANS. Fig. 4 and 5 should provide some results of two out of five (5?) features used (distance and std dev?). Table 2 reports metrics with value and range with no descritpion of what they are and the relevance. The value, for example , is given in pure numbers from 2 cases of the order of unity to a case of the order of 10^6 and then a percentage with ranges not compatible among them. Table 1 is very misterious - not references in the text. It seems that the hardware used by the authors is reported. Table caption is missing.

Conclusion and future work. The authors claim that the developed system emerges as a versatile option for monitoring cryogenic system specifically applied to radio astronomy. I simply do not see how this can be claimed.

This paper needs to be completely re-written if the authors want to address the radio astronomy community. A full description of the working of a G-M cold head should be written as introduction so that the reader know what signals will be analyzed. Then a description of the hardware used: both the cold heads and the hall sensor boards. Then a descritpion in simple terms of the software used with full discussion of all the terms used. Comparison of their software with other software should then be reported and finally the results of the real time analysis of systems. For example - from fig. 4 I understand that a normal cold head has a slightly higher std dev than a system needing maintenance. Why is that? The compressor fault data at zero std dev does not make sense. Perhaps the authors are plotting the amplitudes of the 3 phases? Why only one phase is shown? What about the phase differences? Have they been analyzed? There are many other questions that are potentially interesting and I invite the authors to make a list and address each and every item of the list.

As it is I do not recommend publication unless a major revision is conducted.

Comments on the Quality of English Language

English is OK. I prefer to work on the revised version - if produced.

Author Response

Thank you for your review. Below, I address your corrections point by point, hoping to be thorough enough. It's important to note that the article is classified as "communication" and represents only the first part of a larger project.

  • Although there's a subtle difference between the two devices, we opted to replace the term "cryopump" with "cold head," as the latter is more appropriate in this context.
  • We chose this type of introduction because we believe it's necessary to provide the reader with an understanding of the importance of cryogenics in radio astronomy and the associated basic concepts. This helps introduce the problem to the reader and underscores the need for adopting predictive maintenance in such a system. In this project phase, we focused on monitoring a single parameter as a first step. Currently, we're developing a system that also considers other parameters, such as temperature, vibration, phase shift, and so on.
  • While other applications may not be directly related to predictive maintenance, we mentioned them to introduce the reader to the various utilities of such systems. We adopted this approach considering that the article may be read by individuals without specific experience in this type of system.
  • We have revised the text section describing Figure 1 to better explain the current flow from the compressor to the cold head.
  • We have provided some additional details about the software, a Python-written graphical interface that displays sensor readings and plots the sinusoids of the 3 currents. Three signals are received, one for each phase. However, to simplify the explanation, the process is detailed only for one phase. This information has been integrated into the article. Furthermore, we have enriched the description of Table 1, providing an in-depth explanation of how and why the data was acquired.
  • The metrics listed in Table 2 are commonly used in clustering and aim to compare two similar systems to evaluate their quality. To ensure better reader comprehension, we have included some relevant bibliographic references.
  • We have included the datasheet of the Hall effect sensors used. 
  • Since our project focuses on measuring the currents of an electric motor, we chose not to include technical details on the operation of the cold head, as we believe its internal structure is irrelevant to our measurements. The scenario of the compressor being off is important to ensure that the system issues an alert even in such a situation. While the phase shift of the sinusoids was not considered in this phase, we believe it is an important aspect and are working to integrate it into our system.
  • We are working on an AI-explainable version of the system. At this point, we think that it's not crucial to have a physical explanation for why the features behave in a certain way. The important thing is that the clusters are well differentiated and identifiable by the algorithm.

Please note that modifications to the text are highlighted in bold. Image captions have been improved, and syntax errors have been corrected. Additionally, more details have been added to the conclusions.

Round 2

Reviewer 2 Report

Comments and Suggestions for Authors

The paper has marginally improved. I maintain my doubts about the introduction being not relevant, as written, to the rest of the paper. I think that the paper will be more useful to the readers if the introduction will describe the need to cool HEMT transistors to reduce their noise. More adequate would be, for example, to show the equivalent noise circuit and briefly discuss the noise mechanisms. A plot of the linear trend of RF noise of a typical LNA versus physical temperature would clarify to the reader the problem - including the well known flattening of the noise around 20K physical temperature thus justifying the use of 10K G-M coolers. In addition, the introduction should also include a brief simple description of "incremental clustering" because I suspect that not all the RF engineers/scientists involved in radio astronomy receiver maintenance are familiar with it.

Author Response

Thank you once more for dedicating your time to reviewing our manuscript. In response to your suggestion, we have included a concise explanation of why transistors should be cooled to mitigate noise, supported by a plot from literature and with additional bibliographic references. Additionally, in Section 2.3, we have introduced a brief overview of incremental clustering, along with a reference to a relevant book for further information.